# Liver X Receptor Activation Attenuates Oxysterol-Induced Inflammatory Responses in Fetoplacental Endothelial Cells

**DOI:** 10.3390/cells12081186

**Published:** 2023-04-19

**Authors:** Meekha George, Magdalena Lang, Chaitanya Chakravarthi Gali, Joshua Adekunle Babalola, Carmen Tam-Amersdorfer, Anika Stracke, Herbert Strobl, Robert Zimmermann, Ute Panzenboeck, Christian Wadsack

**Affiliations:** 1Department of Obstetrics and Gynecology, Medical University of Graz, 8036 Graz, Austria; 2Otto-Loewi Research Center for Vascular Biology, Immunology and Inflammation, Division of Immunology, Medical University of Graz, 8010 Graz, Austria; 3My Pura Vida Wellness Private Limited, Hyderabad 500081, India; 4Diagnostics & Research Institute of Pathology, Medical University of Graz, 8010 Graz, Austria; 5Institute for Molecular Biosciences, University of Graz, 8010 Graz, Austria; 6BioTech-Med, 8010 Graz, Austria

**Keywords:** oxysterols, liver X receptors, ATP-binding cassette transporter 1, toll-like receptor 4, placenta, endothelial cells

## Abstract

Oxysterols are oxidized cholesterol derivatives whose systemic levels are found elevated in pregnancy disorders such as gestational diabetes mellitus (GDM). Oxysterols act through various cellular receptors and serve as a key metabolic signal, coordinating inflammation. GDM is a condition of low-grade chronic inflammation accompanied by altered inflammatory profiles in the mother, placenta and fetus. Higher levels of two oxysterols, namely 7-ketocholesterol (7-ketoC) and 7β-hydroxycholesterol (7β-OHC), were observed in fetoplacental endothelial cells (fpEC) and cord blood of GDM offspring. In this study, we tested the effects of 7-ketoC and 7β-OHC on inflammation and investigated the underlying mechanisms involved. Primary fpEC in culture treated with 7-ketoC or 7β-OHC, induced the activation of mitogen-activated protein kinase (MAPK) and nuclear factor kappa B (NFκB) signaling, which resulted in the expression of pro-inflammatory cytokines (IL-6, IL-8) and intercellular cell adhesion molecule-1 (ICAM-1). Liver-X receptor (LXR) activation is known to repress inflammation. Treatment with LXR synthetic agonist T0901317 dampened oxysterol-induced inflammatory responses. Probucol, an inhibitor of LXR target gene ATP-binding cassette transporter A-1 (ABCA-1), antagonized the protective effects of T0901317, suggesting a potential involvement of ABCA-1 in LXR-mediated repression of inflammatory signaling in fpEC. TLR-4 inhibitor Tak-242 attenuated pro-inflammatory signaling induced by oxysterols downstream of the TLR-4 inflammatory signaling cascade. Taken together, our findings suggest that 7-ketoC and 7β-OHC contribute to placental inflammation through the activation of TLR-4. Pharmacologic activation of LXR in fpEC decelerates its shift to a pro-inflammatory phenotype in the presence of oxysterols.

## 1. Introduction

Gestational diabetes (GDM) is a pregnancy complication often associated with maternal and placental inflammation [1,2]. It is well accepted that hyperglycemia leads to the expression of higher serum levels of pro-inflammatory cytokines such as interleukin-6 (IL-6), IL-8, IL-1β, and tumor necrosis factor-a (TNF-α) in women with GDM compared to normal pregnancies [3,4,5]. Cytokines, which are mainly produced by immune cells, are shown to be synthesized in the human placenta, contributing to the low-grade inflammatory condition by transmitting adverse signals to the fetus [6,7]. Hence, if left untreated, pregnancy-related metabolic inflammation could contribute to severe maternal and fetal outcomes such as the development of type 2 diabetes and other metabolic diseases in both mother and the child in their later stages of life [8].

Liver-X receptor (LXR) is a member of the nuclear receptor family of ligand-activated transcription factors known to be involved in the regulation of various cellular functions including cholesterol, fatty acid and glucose homeostasis. LXRs can form heterodimers with retinoid X receptors (RXR), and binding of agonists to either monomer can activate the LXR/RXR complex [9]. Two of the main LXR target genes, ATP-binding cassette transporters (ABC)A-1 and ABCG-1, are involved in reverse cholesterol efflux pathways [10]. Besides its role in cholesterol homeostasis, several studies revealed its potent anti-inflammatory effects in a variety of cell types [11,12]. The molecular basis of their anti-inflammatory effects has yet to be understood in mechanistic detail [13]. One proposed mechanism is that LXR as a monomer can be sumoylated and stabilize the repressor complexes present in the promotor sequence of pro-inflammatory gene activator protein 1 (AP-1) and NFκB [14]. Another mechanism is sumoylation independent but ABCA-1 dependent, where increased cholesterol efflux is initiated from lipid raft domains of the plasma membrane as a result of ABCA-1 induction upon LXR activation. These lipid rafts are known to harbor pattern recognition receptors such as toll-like receptors (TLRs) [15,16]. Upon activation of TLR-4, a cascade of inflammatory signaling pathways will be induced. In one pathway, mitogen-activated protein kinase (MAPK) proteins (p38, JNK, ERK) are phosphorylated, which in turn lead to the activation and translocation of nuclear AP-1 followed by initiation of inflammatory gene transcription. In another pathway, the p65 subunit of nuclear factor kappa B (NFκB) is phosphorylated and translocated to the nucleus and promotes inflammatory gene transcription [16]. These two pathways also cross talk with each other [17,18]. ABCA-1 induction alters membrane cholesterol homeostasis and membrane rigidity with possible influences of TLR-4 signal complex assembly at the plasma membrane for effective activation of the inflammatory signaling cascade. Cholesterol-enriched membrane microdomains are essential for inflammatory cell signaling through membrane receptors [19]. Therefore, TLR-4 activation and the downstream signaling cascade are hindered upon loss of plasma membrane cholesterol content [20]. 

Oxysterols are bioactive lipids that act as regulators of lipid metabolism, inflammation and cell viability and are involved in the progression of several diseases [21]. They are generated mainly via chemical reactions involving reactive oxygen species (ROS) (e.g., 7-ketoC, 7β-OHC) and enzymatic reactions involving cytochrome P450 (e.g., 24-hydroxycholesterol, 27-hydroxycholesterol [22]. Oxysterols can bind to various receptors including LXR, C-X-C motif chemokine receptor 2 (CXCR2) and smoothened and can initiate a number of signaling pathways [23]. They are suggested to play a role in inflammatory diseases such as obesity, atherosclerosis, and neuro-inflammatory diseases [24]. Altered glucose homeostasis and hyperglycemia in GDM result in hyperpolarization of mitochondria and overproduction of ROS, driving the intra-uterine environment to an increased oxidative stress state [25,26]. Previous studies from our group reported that as a result of increased oxidative stress in GDM, concentration of 7-ketoC and 7β-OHC are elevated in cord blood and human fetoplacental endothelial cells (fpEC) from pregnancies complicated with GDM compared to healthy counterparts [27]. Therefore, one may predict that elevated concentrations of oxysterols in pregnancy pathologies such as GDM contribute to the development of the inflammatory phenotype commonly observed in those pathologies. 

The placenta plays a central role in mediating inflammation in women with GDM [28]. Endothelial cells are the major participants in the inflammatory reactions. They are the source and target of many inflammatory mediators [29]. Moreover, fpEC efficiently transport cholesterol to the fetal circulation via ABCA-1 and ABCG-1, thereby playing a major role in placental cholesterol homeostasis [30]. Arterial endothelial cells from the placenta predominantly express genes associated with cholesterol homeostasis, such as LXR and scavenger receptor class B type 1 (SR-B1), and angiogenesis, such as vascular endothelial growth factor (VEGF), in comparison with venous endothelial cells [31,32,33].

Placental endothelial cells are likely to be exposed to a higher concentration of oxysterols during pregnancy complications [20]. Studies about the role of oxysterols in inducing placental inflammation are extremely limited. To the best of our knowledge, studies on the effect of oxysterols in endothelial cells reflecting the inflammatory state are lacking. The potential of LXR in reducing adverse inflammatory effects in conditions of placental pathologies is also poorly understood. Therefore, we considered studying the potential of LXR in repressing inflammation. In particular, we were interested in studying the underlying pro-inflammatory signaling pathways at the fetoplacental vasculature in response to oxysterols. To this end, we used a very well-established in vitro model of primary fpEC, isolated from arterial chorionic vessels of healthy term placentas. We also investigated the impact of LXR activation by the non-steroidal synthetic agonist T0901317 (TO) on inflammatory signaling pathways in fpEC.

Our findings suggest that fpEC exert inflammatory responses upon stimulation with oxysterols, irrespective of their LXR activation ability, whereas pharmacologic activation of LXR by TO displayed protective effects against oxysterols by inhibiting the TLR-4 signaling via an ABCA-1-dependent mechanism. Together, our data point to the potential role of a synthetic LXR agonist as a promising therapeutic to treat inflammation in placental pathologies.

## 2. Materials and Methods

### 2.1. Study Population

This study was approved by the ethical committee of the Medical University of Graz, Austria (29–319 ex 16/17), and all study participants gave voluntary detailed consent. The oral glucose tolerance test (OGTT) was conducted at 24 weeks of gestation to diagnose GDM, and subjects with normal OGGT results and without other medical complications during pregnancy such as hypertension were selected for the study. Clinical characteristics of the study subjects are summarized in Appendix A.

### 2.2. Isolation and Culture of Primary Human Fetoplacental Endothelial Cells

Primary fpEC were isolated from chorionic arteries of human term placentas obtained from normal pregnancies following a well-established protocol [32]. First, arterial vessels were dissected from the apical surface of the chorionic plate. Endothelial cells were isolated by perfusion and digestion of selected arteries with Hank’s balanced salt solution for 8 min (HBSS) (Gibco by Life Technologies, Thermo Fisher Scientific, Waltham, MA, USA) containing 0.1 U/mL collagenase, 0.8 U/mL dispase II (Roche, Vienna, Austria), and 10 mg/mL penicillin/streptomycin (Gibco by Life Technologies, Thermo Fisher Scientific, Waltham, MA, USA). The cell suspension was centrifuged (200× *g*, 5 min) and the obtained cell pellet was re-suspended in endothelial cell growth medium (Promocell, Heidelberg Germany) containing supplements and 10% pregnant serum or medium with platelet lysate and plated on 1% gelatin (Sigma Aldrich, St. Louis, Missouri, USA) coated wells of a 12-well plate. Cells were split into one 6-well plate, one 25 cm^2^ flask and finally one 75 cm^2^ flask accordingly when cells were approx. 80% confluent. Identity and purity of fpEC were confirmed by immunocytochemistry staining of specific endothelial markers such as vWF (A0082, Agilent, Santa Clara, CA, USA); CD31 (mon60021, Sanbio BV, Uden, Netherlands); CD90 (DIA100, Dianova, Hamburg, Germany); Actin smooth muscle (M0851, Agilent, Santa Clara, CA, USA); Vimentin (M0725, Agilent, Santa Clara, CA, USA); MsX Fibroblasts (CBL271, Merck, Darmstadt, Germany); and Desmin (M0760, Agilent, Santa Clara, CA, USA). For maintaining culture, primary cells were grown in Promocell MV media with 5% FCS and extra supplements (Endothelial Cell Growth Supplement, Epidermal Growth Factor (recombinant human), Heparin, Hydrocortisone (Promocell, Heidelberg, Germany). Cells split for up to 10 passages were used for experiments.

### 2.3. RNA Isolation, Reverse Transcription and Real-Time Quantitative PCR (RT-qPCR) 

Total RNA of fpEC was isolated using RNeasy Mini kit (Qiagen, Hilden, Germany) according to the manufacturer’s instructions. Cell monolayer was washed using PBS, and 0.7 mL of QIAzol lysis reagent (Qiagen, Germantown, MD, USA) was added to lyse the cells, and RNA was isolated using the kit. The concentration and the purity of obtained RNA was quantified by measuring 260/280 ratio using the Scandrop 250 (Analytik Jena Jena, Germany), and RNA integrity was examined by gel electrophoresis. cDNA was made using Luna universal reverse transcriptase PCR kit (New England Biolabs Frankfurt, Germany). Real-time quantitative PCR (qPCR) was carried out by using Luna universal qPCR reagent (New England Biolabs, Frankfurt, Germany) in CFX384 or CFX96 real time PCR cycler (Bio-Rad Laboratories, Vienna, Austria). All primer sequences used in this study are listed in Table 1. All primers were designed by crossing the exons to avoid the amplification of genomic DNA. Primer efficiency was verified by standard calibration curves. Gene expression levels were normalized to HPRT1 (Hypoxanthine Phosphoribosyl transferase 1) house-keeping genes, and the results were calculated using the 2^−ΔΔCT^ method. 

### 2.4. Immunoblotting

Monolayers of cultured fpEC were washed twice by ice cold PBS before adding Protein lysis buffer containing protease and phosphatase inhibitor (Roche Diagnostics, Mannheim, Germany). Cell proteins were isolated by vigorous vortexing and sonication of the cells. Protein concentration was quantified by applying bicinchoninic acid (BCA) assay (Thermo Fisher Scientific, Waltham, MA, USA). The loading sample was prepared by mixing equal concentrations of proteins from various treated and non-treated samples in XT loading dye and reducing agent to achieve a final concentration 1x (Biorad, Hercules, CA, USA). The loading samples were boiled at 95 °C for 5 min, and the proteins were separated by sodium dodecyl-sulfate polyacrylamide gel electrophoresis (SDS-PAGE) using 4–12% Bis-Tris Midi Gel (Biorad, Hercules, CA, USA). Separated proteins were transferred from the gels onto 0.2 µM nitrocellulose (trans-blot turbo mini nitrocellulose transfer) membranes (BioRad, Hercules, CA, USA). Membranes were blocked in TBST (Tris-buffered saline with Tween20) containing 5% non-fat dry milk (Bio-Rad, Hercules, CA, USA) for 1 h at room temperature. The membranes were probed with primary antibodies against ABCA-1 (1:2000; ab18180, Abcam, Cambridge, UK), p-p42/44 MAPK (1:3000; 9101, Cell Signaling, Danvers, MA, USA), p42/44 MAPK (1:3000; 9102, Cell Signaling Technology, Danvers, MA, USA), p-JNK (1:2000; 9251, Cell Signaling Technology, Danvers, MA, USA), JNK (1:2000; 9252, Cell Signaling Technology, Danvers, MA, USA), p-p38 MAPK, (1:2000; 9211 Cell Signaling Technology, Danvers, MA, USA), p38-MAPK (1:2000; 9212, Cell Signaling Technology, Danvers, MA, USA), p-p65 NFκB (1:2000; 3033, Cell Signaling Technology, Danvers, MA, USA), p65 NFκB (1:2000; 8242, Cell Signaling Technology, Danvers, MA, USA), ICAM-1 (1:2000; ab109361, Abcam, Cambridge, UK), VCAM-1 (1:2000; ab98954, Abcam, Cambridge, UK), anti-β-actin (1:2000; 4970, Cell Signaling Technology, Danvers, MA, USA) and anti-α-tubulin (1:2000; 2125, Cell Signaling Technology, Danvers, MA, USA) overnight at 4 °C. The membranes were washed with tris-buffered saline with 0.1% Tween^®^ 20 detergent (TBST) three times (10 min). HRP-conjugated secondary antibodies, goat-anti-rabbit HRP-IgG (1:5000, Biorad, Hercules, CA, USA), and horse-anti-mouse HRP-IgG (1:5000, Cell Signaling Technology, Danvers, MA, USA) were used and incubated for 1 h at room temperature. Signals were visualized using the enhanced chemiluminescence (ECL) development method. After the washing steps with TBST, ECL (Bio-Rad, Hercules, CA, USA) was added onto the membranes and incubated for 5 min to develop chemiluminescent signals on the membranes. The chemiluminescent signals were detected, imaged and quantified by using a ChemiDoc system (Bio-Rad, Hercules, CA, USA) and Image Lab software (version 5.2.1, Bio-Rad, Hercules, CA, USA), respectively.

### 2.5. Human Cytokine Multiplex Assay

Procartaplex multiplex Immunoassay kit was purchased from Thermo Fisher Scientific Waltham, MA, USA to determine and quantify released cytokines in cell culture supernatants. Concentration of IL-6, IL-8, IL-1α, TNF-α protein in the cell culture supernatant were measured after treatment of fpEC with oxysterols according to the user’s manual. The BioPlex-200 suspension array system was used to measure the fluorescence intensity of the samples (Biorad, Hercules, CA, USA). All obtained cytokine concentrations in the supernatant were normalized to total cell protein concentration, which was determined by BCA protein quantification assay.

### 2.6. Flow Cytometry

Flow cytometry experiments were performed on an LSR Fortessa flow cytometer (BD Biosciences, Franklin Lakes, NJ, USA). All data were analyzed using the DIVA (BD Biosciences, Franklin Lakes, NJ, USA) and FlowJoTM v10.7.2 software (BD Biosciences, Franklin Lakes, NJ, USA). The fpEC were harvested using accutase (A1110501; Gibco, Thermo Fisher Scientific, Waltham, MA, USA) to detach the cells and resuspended in 40 μL of phosphate buffered saline (PBS). Fc-receptors were blocked by incubating for 10 min with Fc-blockers (Biolegend, San Diego, CA, USA) on ice before staining with antibodies. Fluorescein isothiocyanate (FITC) conjugated anti-human CD-54 (ICAM-1) (353107; 1:40, Biolegend, San Diego, CA, USA) antibody and Allophycocyanin (APC) conjugated anti-human CD106 (VCAM-1) antibody (305809; 1:40, Biolegend, San Diego, CA, USA) were added in right dilution to the cell suspension and incubated for 45 min at 4 °C. Cells were then washed and examined with the cytometer. Dead cells were excluded from analysis using 7-Amino Actinomycin D (7-AAD) (420403; Biolegend, San Diego, CA, USA) cell viability dye that binds specifically to dead cells. Doublet exclusion was performed by plotting the height or width against the area for forward scatter or side scatter. Isotype controls for FITC Mouse IgG1 k (400107, Biolegend, San Diego, CA, USA) and APC mouse IgG1 k (555751, CiteAb, Bath, England) were used as negative control to help differentiate non-specific background signal from specific antibody signal.

### 2.7. LDH Cytotoxicity Assay

The presence of lactate dehydrogenase (LDH) in the cell culture supernatant was quantified using CyQUANT™ LDH Cytotoxicity Assay kit (Thermo Fisher Scientific, Waltham, MA, USA) following the manufacturer’s protocol. Briefly, 20,000 cells were seeded in 96-well plate and grown for 24 h. The cells were then treated with different concentrations of 7-ketoC and 7β-OHC (1, 5, 10, 20, 50 µM) for 24 h. The supernatant was collected and analyzed for the quantification of LDH using the colorimetric method. The absorbance was measured at 490 nm and 680 nm. LDH activity was calculated by subtracting the background absorbance at 680 nm from absorbance at 490 nm.

### 2.8. Statistical Analysis

Experiments were performed in technical replicates of multiple cell preparations (n, biological replicates as depicted in the figure legends). Data are presented as mean ± SEM. For more than two groups, depending on the number of variables, one-way ANOVA or two-way ANOVA followed by Dunnett’s or Tukey’s post hoc test were used to analyze significant differences between groups, using GraphPad Prism 8.3.0 (GraphPad Software, Inc., La Jolla, CA, USA). Values of *p* < 0.05 were considered statistically significant.

## 3. Results

### 3.1. 7-KetoC and 7β-OHC Activate Pro-Inflammatory TLR-4 Signaling in fpEC

Oxysterols are bioactive cholesterol derivatives known to have pro-inflammatory properties in a variety of cell types [20,34]. To investigate if oxysterols elicit inflammation in placental endothelial cells, we treated fpEC with pre-titered non-toxic (obtained from LDH cytotoxicity assay), (Appendix A) concentration of 10 µM of 7-ketoC and 7β-OHC for 6 h and assayed for phosphorylation of MAPK family members and p-65 NFκB, both acting downstream of TLR-4 activation. Both oxysterols strikingly amplified the phosphorylation and hence, the activation of MAPK (JNK, ERK, p38) and p65- NFκB proteins (Figure 1).

### 3.2. 7-KetoC and 7β-OHC Induce Transcription of Pro-Inflammatory Cytokines but Lead to Unchanged Release

To elucidate the pro-inflammatory effects of 7-ketoC and 7β-OHC in fpEC, the cells were treated with 10 µM of 7-ketoC or 7β-OHC for 24 h for RNA isolation, and the expression levels of IL-6 and IL-8 mRNA were measured by RT-qPCR. Among several other measured cytokines whose levels were found elevated in pregnancy pathologies (IL-2, IL-1α, IL-1β, TNF-α), only IL-6 and IL-8 mRNA expression were notable in the basal state and after oxysterol incubation in fpEC. All other analyzed cytokine mRNA expression levels were conversely negligible in fpEC even after LPS stimulation, indicated by its really high Ct values in qPCR (data not shown). Hence, we continued studying only IL-6 and IL-8 expression in association with oxysterol stimulation. We found that IL-6 (Figure 2A) and IL-8 (Figure 2B) mRNA expression were significantly upregulated upon treatment with both oxysterols. Next, the secretion levels of these cytokines were measured in the cell culture supernatant with multiplex assay. Endothelial cells were treated with 10 µM of oxysterols, and supernatants were collected after different time points (3, 6, 12, 24 h). Surprisingly, neither IL-6 (Figure 2C) nor IL-8 (Figure 2D) secretion levels were significantly increased upon oxysterol stimulation compared to the vehicle. We also measured IL-1α and TNF-α secretion after incubation with oxysterols as well as with LPS. Even after LPS stimulation, the concentrations of IL-1α and TNF-α were less than 10 pg/mL (data not shown). Only IL6 and IL8 levels were prominent. Although maximum levels were observed at 12 h, IL6 and IL8 levels did not differ significantly between vehicle and oxysterol treated group. Moreover, in response to LPS after 6 h, significant changes in cytokine levels were observed compared to the vehicle. Hence, the elevated mRNA levels for IL-6 and IL-8 in response to oxysterols did not translate into proteins, indicating posttranslational regulation of IL-6 and IL-8 protein synthesis.

### 3.3. ICAM-1 Expression Is Elevated in fpEC in Response to 7-KetoC and 7β-OHC

Intercellular adhesion molecules-1 (ICAM-1) and (VCAM1) are critically involved in inflammatory responses, enabling binding and trans-endothelial migration of leukocytes [35]. Their cell surface expression is normally upregulated in response to inflammatory stimuli [36]. Therefore, we aimed to check whether oxysterols act as inducers of ICAM-1 and VCAM-1 expression in fpEC. ICAM-1 but not VCAM-1 mRNA was induced by 7-ketoC or 7β-OHC, with expression levels substantially lower compared to LPS (Figure 3A,B). Consistently, only ICAM-1 total cellular expression and cell-surface localization were significantly induced by 7-ketoC or 7β-OHC stimulation in western blot analysis of total cell lysates (Figure 3C) and FACS analysis (Figure 3D,E), respectively.

### 3.4. LXR Activation by Synthetic Agonist T0901317 Inhibits Inflammatory Signaling

LXR activation has been shown to repress pro-inflammatory signaling pathways [13,37]. In pre-clinical animal models, synthetic LXR agonists were shown to ameliorate inflammatory disorders [38]. Several oxysterols are endogenous agonists for LXR [39]. Although 7-ketoC and 7β-OHC bind to LXR, which contributes to slight induction of its target genes (ABCA-1 and ABCG-1) [27], the net result in fpEC is the prevalence of inflammation. Therefore, here we hypothesized that pharmacologic activation of LXR by a high-affinity synthetic agonist may attenuate oxysterol-induced pro-inflammatory responses in fpEC. To test this hypothesis, we pre-incubated fpEC with 2 µM of T0901317 (TO) for 16 h, followed by the addition of oxysterols (7-ketoC, 7β-OHC). LXR activation was confirmed by ABCA-1 induction (Figure 4A). Interestingly TO diminished the oxysterol-stimulated phosphorylation and activation of MAPK and p65- NFκB signaling (Figure 4A). Vehicle was normalized to one (not indicated in the graph). Oxysterol-stimulated ICAM-1 cell surface expression was also diminished in LXR activated cells (Figure 4B,C). TO also attenuated the oxysterol-induced cytokines (IL-6, IL-8) and ICAM-1 mRNA expression and ICAM-1 protein induction in fpEC (Appendix A).

### 3.5. ABCA-1 Induction Is Crucial for LXR-Mediated Repression of Inflammatory Signaling

The transporter proteins ABCA-1 and ABCG-1 induced by LXR are major regulators of cholesterol and phospholipid homeostasis. Moreover, several studies show that reduction in membrane cholesterol dampens TLR activation in lipid raft domains and thereby suppresses inflammation in different cell types [37]. ABCA-1, not ABCG-1, was identified as the major candidate in LXR-mediated repression of inflammatory signaling [12]. Furthermore, basal and LXR-induced expression of ABCA-1 protein was several folds higher than that of ABCG-1 in fpEC [27]. We observed only a non-significant increase of 1.5-fold in ABCG-1 protein after TO treatment, although there was significant ABCG-1 mRNA induction from six biological replicates (data not shown). ABCA-1 mRNA and protein levels after TO treatment were significantly higher (~20 fold) compared to vehicle in all of the biological replicates tested, as observed previously [27]. Therefore, we hypothesized that ABCA-1 was the major candidate gene in TO-mediated inflammatory suppression observed in fpEC. To elucidate the critical involvement of ABCA-1, we used probucol to inhibit the cholesterol efflux activity of ABCA-1. Probucol was originally designed for the treatment of hypercholesterolemia. It is a unique diphenolic antilipidemic compound shown to inhibit the activity of ABCA-1 [40,41]. Supporting evidence showed that probucol is an effective inhibitor of ABCA-1-mediated cholesterol efflux [30]. Inhibition of ABCA-1 translocation to the plasma membrane may partly explain the reduction in cholesterol efflux in the presence of probucol [42]. Inactivation of ABCA-1 in the plasma membrane has been shown in human fibroblast cells with probucol [43]. It diminished basal and LXR-activated apoA-I–dependent cholesterol release from fpEC [30]. Having known the role of probucol in inhibiting LXR-mediated cholesterol efflux, we examined whether probucol plays a role in antagonizing TO-mediated suppression of the oxysterol-induced inflammatory responses. Cells were treated with 10 µM of probucol along with 2 µM of TO for 16 h, followed by oxysterol treatment. As anticipated, probucol antagonized the anti-inflammatory effects of TO indicated by amplification of MAPK and NFκB phosphorylation (Figure 5A) and ICAM-1 cell surface localization (Figure 5B,C). These results were similar to the inflammatory effects induced by oxysterols alone in fpEC. This observation was possibly because of the inhibition of ABCA-1 mediated cholesterol efflux from the plasma membrane stabilizing the effective TLR activation by oxysterols. Similar results were obtained also when measuring mRNA levels of cytokines as well as ICAM-1 and total cellular ICAM-1 protein in the presence of probucol (Appendix A). These results from our study further elucidated the potential anti-inflammatory role of ABCA-1.

### 3.6. 7-KetoC and 7β-OHC Exert Inflammatory Responses in fpEC via TLR-4 Dependent Mechanisms

MAPK and NFκB pro-inflammatory signaling cascades are initiated upon TLR-4 activation [16]. In placental trophoblast cells, 7-ketoC elicits inflammatory responses via TLR-4 mediated mechanisms [20]. The mode of action of 7β-OHC is unknown so far in placental cells. We examined whether the observed oxysterol-mediated enhancement of pro-inflammatory signaling is TLR-4 dependent. To elucidate it, we pre-incubated the cells with TLR-4 specific inhibitor Tak-242 for 2 h, followed by oxysterol stimulation. Cell-permeable Tak-242 acts by disrupting the TLR-4 interaction with the adapter molecules TIRAP and TRAM by selectively binding to TLR-4 via the intracellular Cys747 residue of TLR-4, but not to TLR1–3 or TLR5–10 [44]. Cells were harvested after 6 h for western blotting and after 24 h for qPCR and FACS staining. Interestingly, oxysterol-mediated phosphorylation of p65-NFκB and MAPK proteins and ICAM-1 surface expression were diminished by Tak-242 (Figure 6). Furthermore, Tak-242 inhibited oxysterol-mediated cytokine and ICAM-1 mRNA induction (Appendix A). Ultrapure LPS, which binds to only TLR-4, was used as positive control, and the inflammatory effects of LPS were nullified with Tak-242 (Appendix A). Our results supported the idea of TLR-4 mediated pro-inflammatory gene induction by oxysterols 7-ketoC and 7β-OHC.

## 4. Discussion

In pregnancy disorders hallmarked by inflammation, such as gestational diabetes and preeclampsia, elevated levels of oxysterols have been identified [45,46]. Oxysterols, the endogenous agonists for LXR, have been shown to elicit inflammatory responses in endothelial cells via LXR-independent mechanisms [47]. In particular, exposure to oxysterols evoked inflammatory reactions in placental trophoblasts and impaired their invasion properties [20,48]. Therefore, we aimed to elucidate the interplay between pro- and anti-inflammatory effects of oxysterols in fpEC, which represent the barrier between the placenta and fetus. These particular cells maintain phenotypic heterogeneity in vitro up to 12 passages without the apparent loss of functional and morphological integrity [32]. In this study, we demonstrated for the first time that 7-ketoC and 7β-OHC induce inflammatory responses in fpEC by the activation of the innate immune system cascade. Of note, upon activation of the TLR-4 cascade in fpEC, inflammatory cytokine (IL-6, IL-8) mRNA and ICAM-1 transcription and subsequent protein expression were induced. Pharmacologic activation of LXR in fpEC, by using a non-steroidal synthetic agonist TO, exerted protective effects against oxysterol-induced initiation of inflammatory responses by dampening MAPK (p-38, JNK, ERK) and NFκB phosphorylation. Subsequently, cytokine transcription and ICAM-1 total protein expression were lowered. Probucol, a well-described inhibitor of ABCA-1 efflux activity, reversed the anti-inflammatory effects of TO in fpEC. We also demonstrated that 7-ketoC and 7β-OHC initiate inflammatory responses by activating TLR-4 signaling, and its inhibition proficiently ameliorates the inflammatory signaling induced in fpEC.

In general, endothelial cells are the master regulators of inflammation by providing a steady anti-inflammatory state. However, upon encountering inflammatory stimuli, these cells upregulate cell adhesion molecules on their cell surfaces, enabling leukocyte binding and trans-endothelial migration. Moreover, endothelial cells produce chemokines and cytokines to counteract adverse effects on the vasculature [29,49]. Furthermore, if the inflammatory environment persists throughout the pregnancy period without being resolved, this may contribute to the development of cardiovascular and metabolic diseases in the mother and child later in their life [3]. Therefore, it has become most urgent to understand inflammatory processes happening at the feto–maternal interface during gestation. Placental endothelial cells have been known to play a crucial role in cholesterol homeostasis in conditions of GDM [27]. The fetal blood is in direct contact with fpEC [50]. Moreover, fpEC from GDM pregnancies overexpress LXR target genes such as ABCA-1 and ABCG-1 to efficiently efflux excess cholesterol to HDL in order to maintain cholesterol homeostasis [27]. Therefore, fpEC may serve as a target for cellular inflammatory actions as a result of cholesterol accumulation in the human placenta [51].

Previous studies from our lab reported significantly higher levels of ROS in GDM fpEC. This was corroborated by elevated levels of 7-ketoC and 7β-OHC species in cord blood and fpEC from GDM compared to normal pregnancies [27]. Several clinical studies reported the levels of plasma oxysterols in the range of 30–40 ng/mL in normal conditions [52,53]. However, the oxysterol concentration detected in atherosclerotic lesions is found to be increased by several folds. The concentration of oxysterols in atherosclerotic plaques can vary depending on a number of factors, including the severity of the disease and the stage of the plaque [54,55]. GDM and predominantly preeclamptic placentas appear often together with atherosclerotic lesions [56,57,58]. Hence, one may predict that the concentration of oxysterols in such focal lesions is much higher than normal plasma levels [20]. We tested the cytotoxic effects of oxysterols on fpEC at different concentration and we decided to use non-toxic 10 µM of oxysterols for all in vitro experiments reflecting a pathophysiological concentration as may occur in placentas of pregnancy pathologies. Additionally, our data on LDH assays nicely correlated with the described cytotoxic effect of oxysterols at higher concentrations of above 20 µM [24].

Both auto-oxidation products of cholesterol (7-ketoC and 7β-OHC) used in our study participated in eliciting inflammation in fpEC. These endogenously formed oxysterols are known for their major cytotoxic and inflammatory properties [59]. Both species profoundly induced transcription of IL-6, IL-8, and ICAM-1 in fpEC. Additionally, they induced amplification of ICAM-1 but not VCAM-1 protein expression and cell surface localization. Surprisingly, induction of IL-6 and IL-8 transcription did not lead to changes in their secretion. However, LPS significantly induced the release of cytokines even after 6 h of incubation, which is in accordance with other, similar studies. It is well known that LPS can activate multiple TLR-4 downstream signaling pathways that eventually lead to the transcription, followed by translation, of cytokines [60]. Taking all of these results together, we contemplated a resilient mechanism that avoids fpEC, although induced by pro-inflammatory stimuli, to continue with inflammatory responses by translation of cytokines. In contrast, there is strong evidence that oxysterols may contribute to the secretion of cytokines in different cell types [34,47]. Moreover, 7-ketoC has been shown to induce IL-6 and TNF-α secretion in primary placental trophoblasts [20]. However, we believe that placental endothelial cells may regulate inflammation more distinctly than trophoblasts do, in order to cope with chronic inflammation as a first-line post-translational defense mechanism against the fetus.

In addition, out of several inflammatory cytokine levels (IL-2, IL-1α, IL-1β, TNF-α) analyzed by qPCR, we observed notable expression of only IL-6 and IL-8 in fpEC, although it is reported that the endothelial cells participate in the production of several pro-inflammatory cytokines [61]. Additionally, we also looked at cell surface expression of E-selectin, whose expression is also negligible (data not shown) on fpEC. We suggest the existence of an inflammatory regulatory mechanism employed by the placental endothelial cells to keep the inflammatory status low as a protective measure to prevent the fetus from being exposed to a persistant inflammatory environment. In normal fpEC, the induction of only ICAM-1 by oxysterol may minimize the inflammatory reactions and subsequent leukocyte infiltration and vascular permeability, which is efficiently performed by the combined activity of ICAM-1, VCAM-1 and E-selectin [62,63]. The post-transcriptional mechanism proposed herein, which prevents IL-6 and IL-8 mRNA from being translated in oxysterol treated fpEC, need to be studied in future. In fpEC from GDM, it has been demonstrated that specific miRNAs lower ICAM-1 expression by post-translational effects compared to normal cells, which is suggested to be a protective placental endothelial cell-specific mechanism to avoid leukocyte transmigration and helps maintain a steady anti-inflammatory state regardless of the inflammatory milieu in conditions of GDM [64].

The concept that alterations of inflammatory markers in the mother may not necessarily be reflected by similar changes in the fetal circulation is, meanwhile, well accepted [3]. Inflammatory mediators such as leptin, IL-6, TNF-α, and IL-10 were increased in GDM mothers, while in their neonates, their levels were found to be decreased [65]. It is, therefore, possible that the placenta, and thereby the fetoplacental endothelium, acts as a sensor responding distinctly to the inflammatory maternal environment in order to maintain normal placental function. We speculate that oxysterols induce a negative feedback loop that could attenuate the early inflammatory response, such as cytokine release through one of its receptors, such as LXR [39].

Apart from the studied pro-inflammatory mediators, the MAPK pathway can give rise to the expression of several other inflammatory factors such as cox-2, prostaglandins, metallomatrix proteases and monocyte chemoattractant proteins [66,67]. We did not access the expression of all possible inflammatory mediators, as it is beyond the scope of this study. Therefore, we predicted that oxysterol elicits several other inflammatory responses downstream of TLR-4 activation in fpEC, and we decided to explore the anti-inflammatory potential of nuclear receptor LXRs, which have been identified to play a role in suppressing inflammation [12,39]. To test the responses of LXR in our fpEC upon oxysterol treatment, we introduced the non-steroidal high-affinity LXR agonist called TO. We observed attenuated NFκB and MAPK phosphorylation, decreased cytokine transcription, and ICAM-1 total and surface protein levels in fpEC pre-treated with TO prior to oxysterol addition. Moreover, TO significantly induced protein expression of one of the LXR target genes, ABCA-1, in fpEC by 20- to 30-fold. ABCG1, which is also an LXR target gene, was induced by TO to merely 1.7-fold in our cells (data not shown). Studies already demonstrated that ABCA-1, not ABCG-1, is the major cholesterol efflux transporter involved in the regulation of inflammation [12]. Therefore, we hypothesized that it is rather ABCA-1 that is involved in LXR-mediated inhibition of inflammatory responses [12]. Evidence suggests that changes in the lipid content of plasma membranes can impair inflammatory signaling [19,68]. ABCA-1 induction increases cholesterol efflux from the plasma membrane, redistributes cholesterol content in the membrane, and alters its fluidity, thereby disrupting TLR-4 activation which requires gathering of various accessory molecules and adapter proteins at the plasma membrane [12]. Moreover, the ability of ABCA-1 to efflux oxysterols efficiently from the cells cannot be fully excluded [69]. Previous investigations demonstrated the ability of ABCA-1 to efficiently remove 25-hydroxycholesterol from cells [70]. It is so far unknown whether 7-ketoC and 7β-OHC are trafficked out of the cells by ABCA-1. However, it has been shown that ABCG-1 promotes the efflux of 7-ketoC in macrophages [71].

To confirm the impending role of ABCA-1 in attenuating inflammation, we pre-incubated the cells with TO along with probucol, a drug that can inhibit the cholesterol efflux mediated by ABCA-1, and its translocation to the plasma membrane, thereby disrupting the cholesterol trafficking from the plasma membrane [42,43]. As anticipated, probucol antagonized the anti-inflammatory effects of TO in fpEC. Therefore, it is clear that ABCA-1 expression and activity are required for LXR-mediated lowering of inflammatory signaling. However, we did not exclude the possibility of trans-repression of inflammatory gene expression by sumoylated LXR in our cells. Taken together, our findings further confirmed the potential of LXR as a target for activation of the resolution cascade of inflammation and complications associated with it. The protective effects of TO we observed in fpEC are comparable with what was observed in placental trophoblast cells previously [20]. Additionally, LXR activation in fpEC was also effective against LPS-induced inflammatory responses (Appendix A), as demonstrated in other cell types previously [12].

Of note, TO is also known to activate farnesoid X receptor (FXR) more potently than a natural bile acid FXR ligand. The EC_50_ of TO for LXR is 50 nm, while the EC_50_ for FXR is 5 µM, which makes TO a much more potent activator of LXR [72]. Therefore, we predicted that the effective activation of FXR by TO was not occurring in fpEC at the given concentration of 2µM. However, we did not rule out the possibility of mild or transient activation of FXR by TO in our cells. FXR acts as a key metabolic regulator and potential drug target for many metabolic syndromes including chronic inflammatory diseases [73]. In our study, we elucidated the crucial role of LXR target gene ABCA-1 in the regulation of inflammation, and we speculated that activation of LXR plays a major role here.

Oxysterols are able to trigger pro-inflammatory signaling via the activation of TLR-4 [20,74]. We here demonstrated that oxysterols elicit inflammation in fpEC via TLR-4 activation. This was confirmed by treating the cells with a TLR-4 specific inhibitor called Tak-242 in our studies. Tak-242 incubation along with oxysterols significantly reduced the phosphorylation levels of NFκB and MAPK proteins and mRNA expression of cytokines. Furthermore, it also diminished ICAM-1 mRNA and total cellular and surface protein expression. The efficiency of Tak-242 was confirmed using ultra-pure LPS as positive control, which could only bind and activate TLR-4. [75]. Tak-242 completely abolished the LPS-induced inflammation in fpEC, while it partly abolished oxysterol-induced response, albeit still significantly. Our data suggest that the observed impacts of oxysterols are mediated by activation of TLR-4 signaling. This result is in line with previous investigations that demonstrated TLR-4 mediated effects of oxysterol in placental trophoblasts [20]. We here do not rule out the possible involvement of other TLRs in oxysterol-mediated inflammation. However, our results indicate a crucial role of TLR-4 in placental inflammation mediated by oxysterols. The pro-inflammatory phenomenon instigated by oxysterols in fpEC represents the net result of the activation of two opposing pathways: one being the activation of TLR-4 MAPK pro-inflammatory signaling (LXR-independent mechanism) and the other pathway being the LXR activation and reduction of membrane cholesterol content as suggested previously [20]. In our outlined study of fpEC, the inflammatory signaling pathway induced by 7-ketoC or 7β-OHC predominated the anti-inflammatory signaling.

The precise mechanism by which oxysterols activate TLR is not fully understood. Based on the literature, it is thought to involve the binding of oxysterols with MD-2, a TLR-4 ancillary molecule, essential for the recognition by TLR-4. It has also been shown that cholesterol can bind to MD2 [76]. Further, oxidized cholesterol ester has been shown to recruit MD2 and induce TLR-4 dimerization [77]. Oxysterol-induced TLR-4 activation may also involve similar mechanisms.

In conclusion, we have shown that pathophysiological concentrations of oxysterols contribute to placental inflammation, a general characteristic of GDM placenta. Placental endothelial cells might have a stringent regulatory mechanism in place to prevent the developing fetus from being exposed to an inflammatory environment. Exogenous triggers such as oxysterols induce inflammatory responses in endothelial cells from placenta. Activation of LXR by a synthetic agonist such as TO could impair the inflammatory signaling through an ABCA-1 dependent mechanism. Taking together, these findings suggest that LXR agonist can be a potential therapeutic to treat inflammation and its consequences associated with GDM.

We also demonstrated that oxysterols, despite their LXR activation ability, induce inflammation at pathophysiological concentrations and play an important role in placental pathologies. Our findings have potentially important implications for the understanding and resolution of placental inflammation occurring during pregnancy. These findings are significant because they suggest that high levels of oxysterols accompanies in pregnancy disorders can contribute to placental inflammation. Additionally, our research supports the role of ABCA-1 in reducing inflammation, which is linked to its well-known cholesterol efflux capacity. There is a need for further research to determine whether oxysterols trigger a feedback mechanism in fpEC that could potentially prevent the translation of cytokines (Figure 7).

## Figures and Tables

**Figure 1 cells-12-01186-f001:**
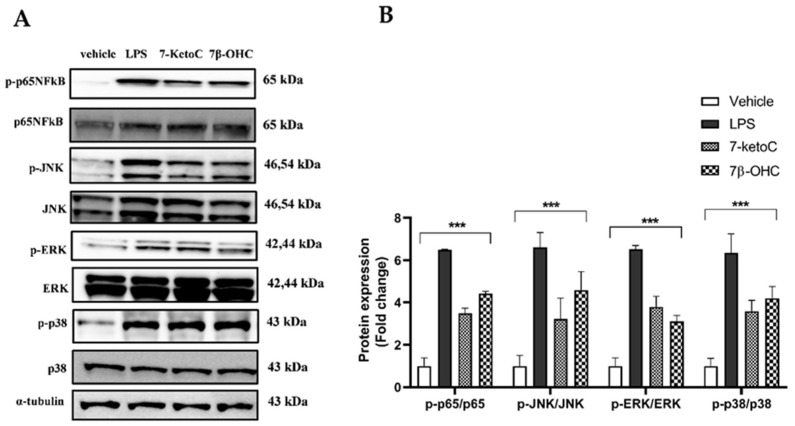
TLR-4 signaling is enhanced in fpEC upon exposure to oxysterols. (**A**) Representative western blot showing phosphorylation of MAPK (p38, JNK, ERK) proteins and NFκB (p65 subunit) in fpEC treated with 7-ketoC or 7β-OHC. LPS (100 ng/µL) treated cells served as the positive control. (**B**) Quantitative analysis of immunoblots (oxysterol-treated cells relative to vehicle control; *n* = 6). Data are presented as mean ± SEM. Statistically significant differences between vehicle and treatments were calculated using two-way ANOVA, followed by Dunnett’s post hoc test. *** *p* < 0.001.

**Figure 2 cells-12-01186-f002:**
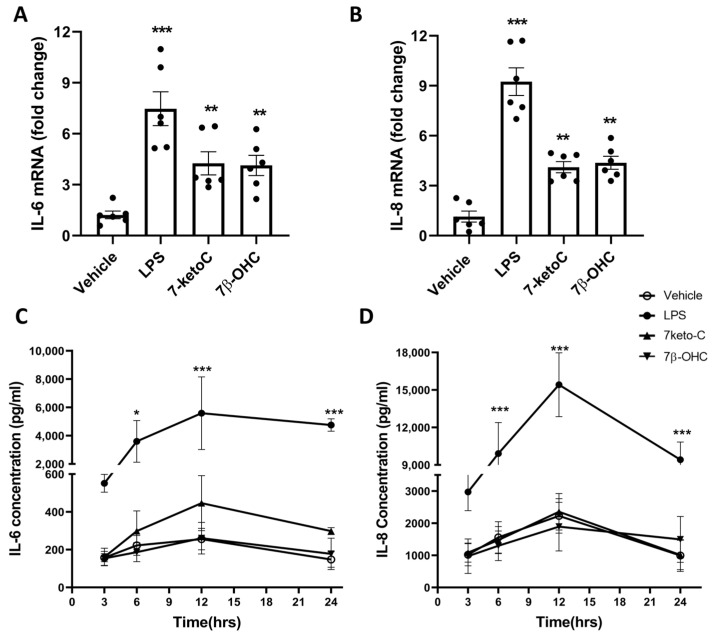
7-KetoC and 7β-OHC induce transcription but not translation of IL-6 and IL-8. IL-6 (**A**) and IL-8 (**B**) mRNA expression are significantly upregulated by both 7-ketoC and 7β-OHC in fpEC (*n* = 6). Fold changes are expressed relative to vehicle controls. LPS (100 ng/uL) used as positive control strikingly amplified cytokine mRNA transcription. Significant differences between the vehicle and oxysterol-treated groups were calculated using one-way ANNOVA, followed by Dunnett’s post hoc test. IL-6 (**C**) and IL-8 (**D**) concentrations in supernatants of cell culture media of fpEC stimulated with oxysterols are not significantly different compared to the vehicle at any time point (*n* = 4). LPS (100 ng/uL) served as positive control. Statistically significant differences between vehicle and treatments at different time points were calculated using two-way ANOVA, followed by Tukey’s post hoc test. Data are presented as mean ± SEM. * *p* < 0.05, ** *p* < 0.01 and *** *p* < 0.001.

**Figure 3 cells-12-01186-f003:**
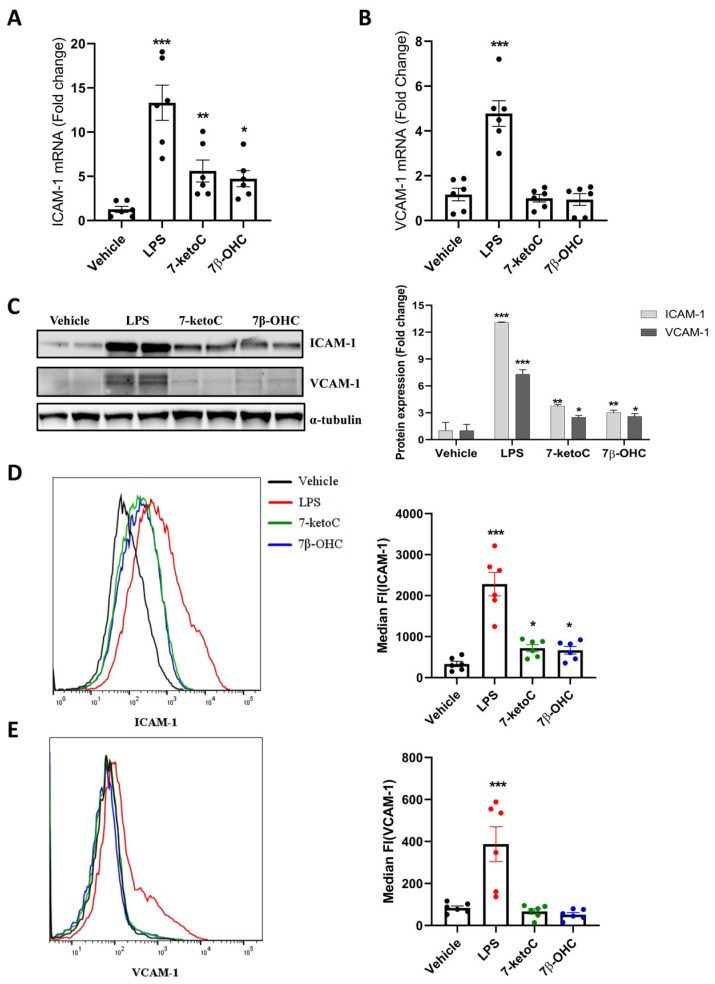
ICAM-1 but not VCAM-1 expression is enhanced by oxysterols. Treatment with 7-ketoC or 7β-OHC (10 µm) for 24 h significantly augments ICAM-1 (**A**) but not VCAM-1 (**B**) mRNA in fpEC (*n* = 6). LPS (100 ng/µL) served as the positive control. (**C**) Total cellular ICAM-1 protein is significantly induced by oxysterols. (**D**) Cell surface staining followed by FACS analysis of oxysterol-treated cells reveals significant induction of ICAM-1. (**E**) LPS but not oxysterols induce surface localization of VCAM-1. Statistically significant differences between vehicle and oxysterol treatment group were calculated using one-way ANOVA, followed by Dunnett’s post hoc test. Data are presented as mean ± SEM * *p* < 0.05, ** *p* < 0.01 and *** *p* < 0.001.

**Figure 4 cells-12-01186-f004:**
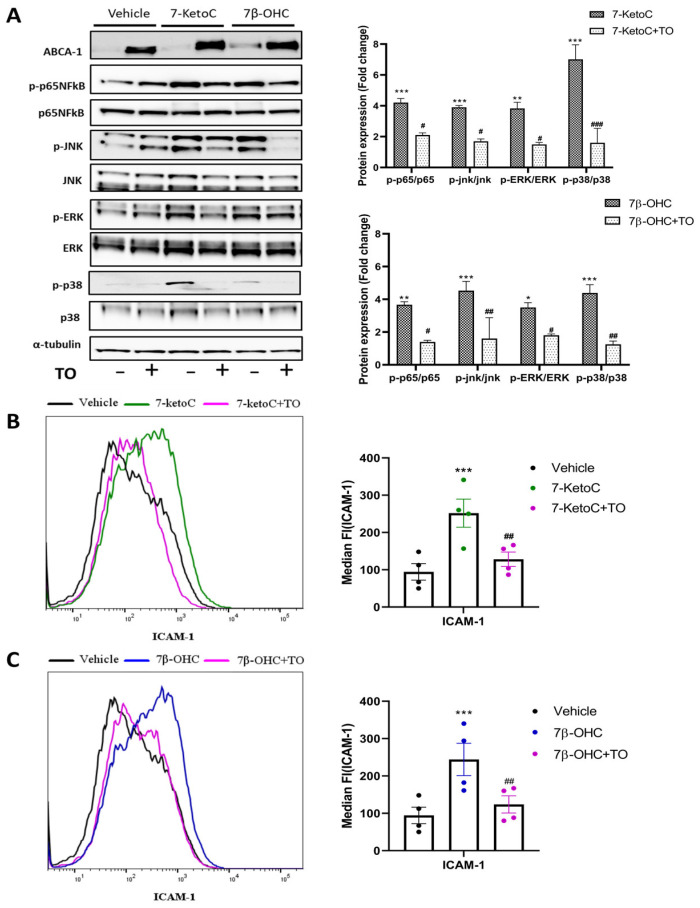
Synthetic LXR agonist TO reduces oxysterol-stimulated inflammatory responses in fpEC. (**A**) Representative western blot image and densitometric analysis demonstrate pronounced anti-inflammatory effects of TO in fpEC (*n* = 5) treated with 7-ketoC or 7β-OHC, confirmed by reduced phospho-p65 NFκB and phospho-MAPK levels in the TO pre-incubated cells prior to oxysterol stimulation. Each phosphorylated protein is normalized to its respective total amount of proteins. The fold expression of the vehicle is normalized to one (not shown in the graphs). The fold change of target protein expression levels after oxysterol stimulation demonstrated is relative to the vehicle control. FACS histogram and respective median fluorescence intensity plot showing decrease in ICAM-1 cell surface expression level with TO pre-incubation followed by 7-ketoC (**B**) or 7β-OHC (**C**) stimulation (*n* = 4). Data are presented as mean ± SEM. Statistically significant differences between vehicle and oxysterol treatment (indicated with *) and between oxysterol and oxysterol + TO treatment (indicated with #) were calculated using two-way ANOVA (for phospho-proteins) and one-way ANOVA (for ICAM-1), followed by Tukey’s post hoc test. #/* *p* < 0.05, ##/** *p* < 0.01, and ###/*** *p* < 0.001.

**Figure 5 cells-12-01186-f005:**
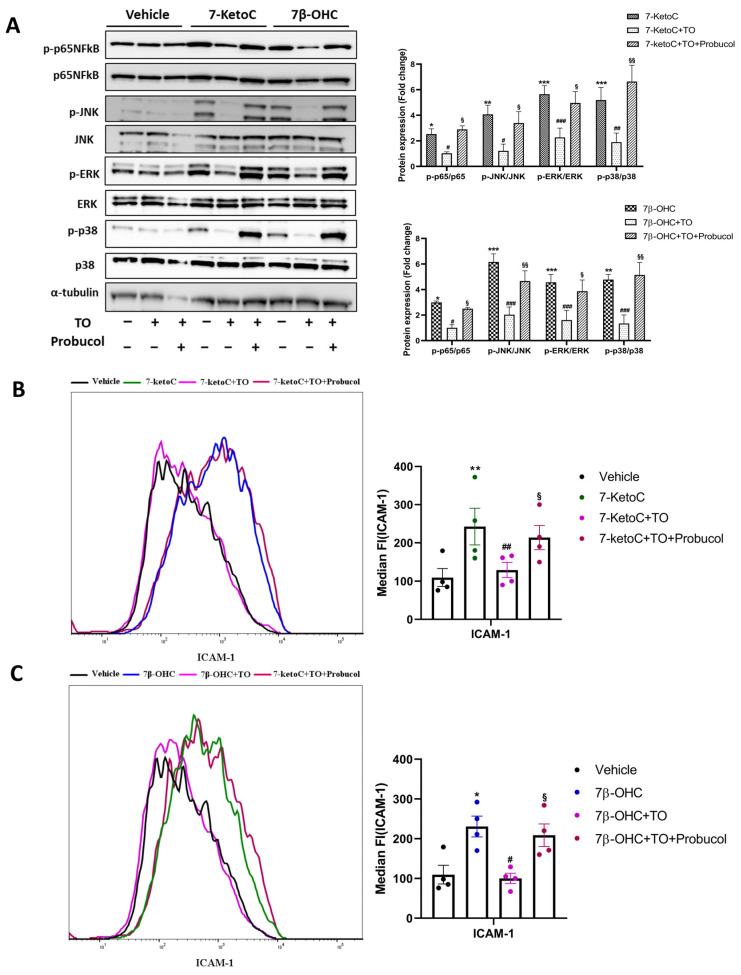
Probucol antagonized the protective effects of TO against oxysterols. (**A**) Representative immunoblot and densitometric analysis after probucol treatment showing MAPK and NFκB signaling re-activation in fpEC (*n* = 4), as created in oxysterol-treated cells. Expression of all targets in vehicle control is normalized to one (not shown in the graph), and target protein expression in treatment groups demonstrated are relative to the vehicle. (**B**,**C**) Cell surface expression of ICAM-1 shows the same pattern in cells treated with probucol (*n* = 4). Data are presented as mean ± SEM. Statistically significant differences between vehicle and oxysterol treatment (indicated with *), between oxysterol and oxysterol + TO treatment (indicated with #), and between oxysterol + TO and oxysterol + TO + probucol (indicated with §), were calculated using two-way ANOVA for phospho-proteins) and one-way ANOVA (for ICAM-1), followed by Tukey’s post hoc test. §/#/* *p* < 0.05, §§/##/** *p* < 0.01 and ###/*** *p* < 0.001.

**Figure 6 cells-12-01186-f006:**
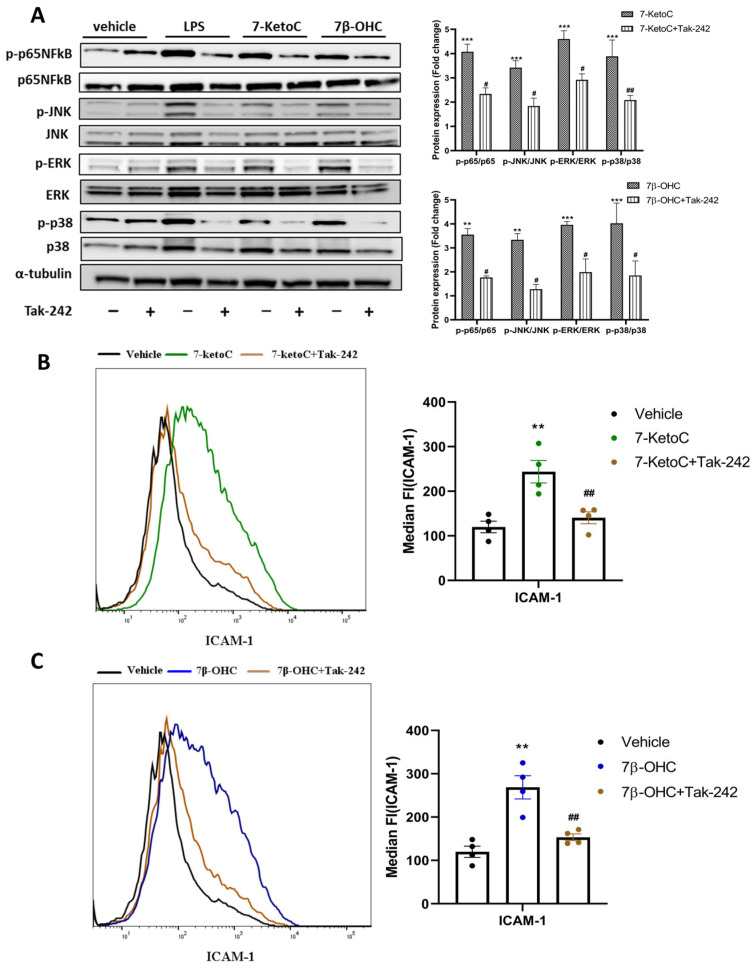
TLR-4 inhibition by Tak-242 ameliorates oxysterol-induced inflammatory responses in fpEC. (**A**) Representative western blot and densitometric analysis showing decreased activation of MAPK proteins, p-65 NFκB and expression of ICAM-1 in oxysterol-treated fpEC by TLR-4 inhibition (*n* = 5) using Tak-242. Target gene fold expression in vehicle is normalized to one (not shown in the graph), and the fold change expression of all targets in treatment groups is relative to the vehicle. (**B**,**C**) Similar results were also obtained for ICAM-1 surface expression in response to Tak-242, as can be seen from the representative FACS histogram and median fluorescence intensity plot (*n* = 4). Data are presented as mean ± SEM. Statistically significant differences between vehicle and oxysterol treatment (indicated with *) and between oxysterol and oxysterol + Tak-242 treatment (indicated with #) were calculated using two-way ANOVA (for phospho-proteins) and one-way ANOVA (for ICAM-1), followed by Tukey’s post hoc test. # *p* < 0.05, ##/**
*p* < 0.01 and *** *p* < 0.001.

**Figure 7 cells-12-01186-f007:**
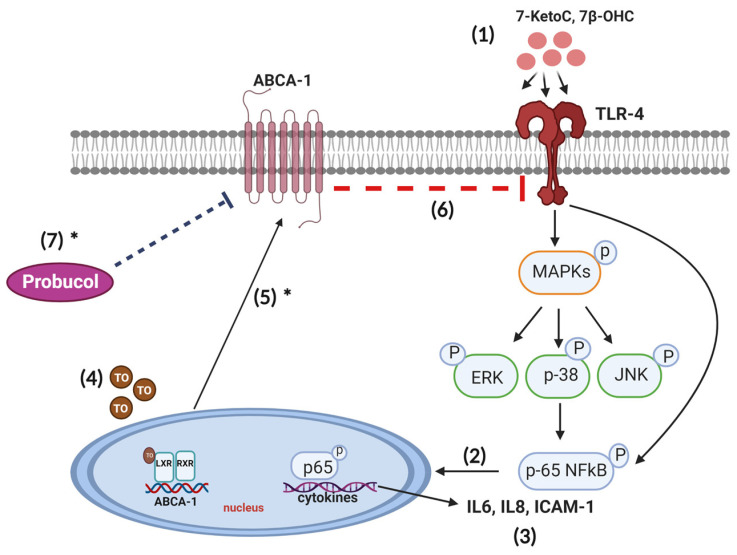
Proposed scheme of ABCA-1 mediated inhibition of TLR-4 activation by oxysterol in human fetoplacental endothelial cells. (1) When fpEC are exposed to 7-ketoC and 7β-OHC, TLR-4 signaling cascade is activated, with concomitant MAPK (JNK, p38, ERK) and p-65 NFκB phosphorylation. (2) Upon phosphorylation, p-65 NFκB complex translocates to the nucleus and (3) initiates transcription of inflammatory mediators such as IL-6, IL-8 and ICAM-1. (4) Treatment with T0901317 activates LXR/RXR heterodimer, which in turn induces ABCA-1 expression and (5) subsequent translocation to the plasma membrane. ABCA-1 initiates membrane cholesterol efflux and alters membrane cholesterol homeostasis [12]. (6) Depletion of cholesterol from lipid rafts of the plasma membrane disrupt TLR-4 activation by oxysterols [20]. (7) Probucol inhibits ABCA-1 mediated cholesterol efflux in fpEC [30], hence favoring oxysterol to activate TLR-4 signaling, while membrane cholesterol is maintained. * already shown in fpEC by others [27,30].

**Table 1 cells-12-01186-t001:** Primer sequences used for RT-qPCR.

Gene	Company	Forward/Reverse Primer
HPRT-1	Thermo Fisher Scientific	F: GACCAGTCAACAGGGGACAT R: CTGCATTGTTTTGCCAGTGT
IL-6	Thermo Fisher Scientific	F: CCACACAGACAGCCACTCAC R: TGCCTCTTTGCTGCTTTCAC
IL-8	Thermo Fisher Scientific	F: GACCACACTGCGCCAACAC R: CTTCTCCACAACCCTCTGCAC
ICAM-1	Thermo Fisher Scientific	F: ATGCCCAGACATCTGTGTCC R: GGGGTCTCTATGCCCAACAA
VCAM-1	Thermo Fisher Scientific	F: GGGAAGATGGTCGTGATCCTT R: TCTGGGGTGGTCTCGATTTTA

## Data Availability

The data generated in the study are presented in the article and Appendix A. The data presented in this study are available on request from the corresponding author.

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
