# Peer review of "Liver X Receptor Activation Attenuates Oxysterol-Induced Inflammatory Responses in Fetoplacental Endothelial Cells"

_cells, 2023, doi:10.3390/cells12081186_

Round 1

Reviewer 1 Report

The effects of 7-ketoC and 7β-OHC on inflammation and the underlying mechanisms involved were investigated in this manuscript. The stimulatory effect of oxysterol on MAPK and NFκB signaling, IL-6 and IL-8 expression levels and ICAM-1 total cellular expression was demonstrated in fetoplacental endothelial cells. Further, it is shown that LXR activation by synthetic agonist TO901317 repressed oxysterols induced stimulation of proinflammatry signaling pathways and ICAM-1 cell surface expression. Using probucol (as inhibitor of LXR target gene ABCA-1) and TLR-4 inhibitor (Tak-242) it is shown that ABCA-1 mediated cholesterol efflux and influence TLR-4 activation by oxysterols.

The attached scheme is very illustrative. In Figure 7 legend the number (7) should be added before the last sentence to correspond with illustration (the only remark).

The results show that the protective effects of the LXR synthetic agonist TO against the oxysterol-induced inflammatory response may be (potentially) significant in therapy when it comes to the treatment of inflammation in placental pathologies.

1. What is the main question addressed by the research?
The mechanism of repression of the inflammatory response caused by oxysterols, which is mediated by the activation of LXR in fetoplacental cells, was investigated.

2. Do you consider the topic original or relevant in the field? Does it
address a specific gap in the field?
Although the mechanism of action of LXR has been extensively studied since 2000 as potential therapeutic application of LXR agonists could be used in numerous disorders, including atherosclerosis, diabetes, cancer, Alzheimer disease the action of oxysterols and the role of LXR in fetoplacental endothelial cells is a novelty. Thus, these results cover the period of gestation, as a very specific period during the life cycle.

3. What does it add to the subject area compared with other published material?
As mentioned above, conditions related to placental and maternal inflammatory disorders and diseases, because fetoplacental endothelial cells are the first line that control the passage of substances and inflammatory signals from the mother to the fetus.

4. What specific improvements should the authors consider regarding the methodology? What further controls should be considered?
None. The material and methods applied are adequate.

5. Are the conclusions consistent with the evidence and arguments presented and do they address the main question posed?
The conclusions are consistent with the presented results. All presented results on the basis of which the main conclusion and proposed and scheme were derived (Figure 7) are given in the Reviewers form.

6. Are the references appropriate?
Yes, the references are appropriate.

7. Please include any additional comments on the tables and figures.
Remark on Figure 7 is alredu done. There is no any additional comments on the tables and figures.

Reviewer 2 Report

My concerns related to this manuscript:

I found a similar article already published in Placenta journal in 2021 by the same author Meekha Geroge et al.

A clarity is required, how this manuscript is novel as compared to the article published in Placenta.

Details of the published article in Placenta:

Liver-X receptor activation attenuates oxysterol induced inflammatory response in human feto-placental endothelial cells in gestational diabetes mellitus
  • September 2021
  • Placenta 112:e37
  • DOI: 
  • 10.1016/j.placenta.2021.07.121

  • Meekha Geroge 
  • Magdalena Lang
  • Carmen Tam-Amersdorfer

Reviewer 3 Report

In this manuscript, George and colleagues present their study on the effects and mechanisms of action of oxysterols on inflammation-related gene expression and signal transduction cascades in fetal placental endothelial cells. Since chronic inflammation associated with hyperglycemia and gestational diabetes can contribute to metabolic conditions, understanding the potential roles of oxysterols and liver X receptors in placental endothelial cells may provide important insights and potential anti-inflammatory mechanisms for therapeutic intervention. While the overall quality of the study and writing are good, there are a number of issues which the authors should consider addressing prior to publication:

1.       The authors chose to treat with 10uM ketoC and 7β-OHC for their study, using cytotoxicity assay results to determine the highest non-toxic concentration. While they did mention that this concentration is achievable in pathological states, what is the evidence for this assertion? Are such concentrations of oxysterols detected in cord blood? On a related note, wouldn’t the concentration used also induce an oxidative stress response?

2.       It is unclear whether the effects observed by the authors are indeed an inflammatory response, especially in comparison to the cytokine responses induced by LPS. TLR activation seems to play a role in the effects of oxysterols on signal transduction cascades but corresponding changes in cytokine production was not detected. ICAM1 protein levels do increase in the endothelial cells, but is this sufficient to trigger or coordinate an inflammatory response, as compared to cytokines?

3.       To demonstrate the potential involvement of LXRs, the authors treated the cells with 10uM of T0901317 (there is a typo in the manuscript where the compound is listed as TO901317, the first number 0 was erroneously replaced with the letter O).  While this Tularik compound is commonly used as a synthetic LXR agonist, at the concentration selected by the authors it can also activate the farnesoid X receptor. The implications for this dual-specificity of the ligands used in the study should be addressed or discussed. Also, if T0901317 or LXR activation is indeed anti-inflammatory, was it effective against the LPS-induced inflammatory response?

4.       The use of TLR4 inhibitor by the authors demonstrate that the observed effects of oxysterols on the activation of signaling molecules and ICAM1 expression are mediated by TLR4. What is the proposed mechanism by which oxysterols can activate TLRs?

5.       The bottom of Figure 5A where treatments with T0901317 and probucol are indicated appears to be mislabeled. The second lane in each set of treatments should only have been treated with T0901317 only and not both as is currently indicated.     

Round 2

Reviewer 2 Report

I am satisfied with the reply of the authors. Can be accepted in present form.